# A New CT Analysis of Abdominal Wall after DIEP Flap Harvesting

**DOI:** 10.3390/diagnostics12030683

**Published:** 2022-03-11

**Authors:** Tito Brambullo, Eva Kohlscheen, Diego Faccio, Francesco Messana, Roberto Vezzaro, Giulia Pranovi, Stefano Masiero, Sandra Zampieri, Barbara Ravara, Franco Bassetto, Vincenzo Vindigni

**Affiliations:** 1Plastic and Reconstructive Surgery Unit, Department of Neurosciences, University of Padua, 35128 Padua, Italy; eva.kohlscheen@aopd.veneto.it (E.K.); facciodiego@outlook.it (D.F.); francesco.messana@aopd.veneto.it (F.M.); franco.bassetto@unipd.it (F.B.); vincenzo.vindigni@unipd.it (V.V.); 2Radiology Unit, Civil Hospital of Pescara, 65124 Pescara, Italy; rvezzaro@gmail.com; 3Rehabilitation Unit, Department of Neurosciences, University of Padua, 35128 Padua, Italy; g.prano.vna@gmail.com (G.P.); stef.masiero@unipd.it (S.M.); 4Department of Biomedical Sciences, University of Padova, 35131 Padova, Italy; sanzamp@unipd.it (S.Z.); barbara.ravara@unipd.it (B.R.); 5Department of Surgery, Oncology and Gastroenterology, 3rd Surgical Clinic, University of Padova, 35128 Padua, Italy; 6Myology Center, University of Padova, 35122 Padua, Italy

**Keywords:** abdominal wall, breast reconstruction, CT, DIEP flap, donor site morbidity, rectus muscle

## Abstract

The abdominal microsurgical flap based on the deep inferior epigastric artery perforator (DIEP) flap has become the most popular option worldwide for autologous breast reconstruction. Several authors have investigated the results of reconstructed breasts, but the literature lacks systematic reviews exploring the donor site of the abdominal wall. To fulfil our aims, a new diagnostic muscle imaging analysis was designed and implemented. This study focused on rectus abdominal muscle morphology and function in a single series of 12 consecutive patients analysed before and after breast reconstruction with a microsurgical DIEP flap. Patients were divided into two groups, namely, “ipsilateral reconstruction” and “contralateral reconstruction”, depending on the side of the flap harvest and breast reconstruction, then evaluated by computed tomography (CT) scans scheduled for tumor staging, and clinically examined by a physiatrist. Numerous alterations in muscle physiology were observed due to surgical dissection of perforator vessels, and rectus muscle distress without functional impairment was a common result. Postoperatively, patients undergoing “contralateral reconstruction” appeared to exhibit fewer rectus muscle alterations. Overall, only three patients were impacted by a long-term deterioration in their quality of life. On the basis of the newly developed and implemented diagnostic approach, we concluded that DIEP microsurgical breast reconstruction is a safe procedure without major complications at the donor site, even if long-term alterations of the rectus muscle are a common finding.

## 1. Introduction

Over the past 20 years, the deep inferior epigastric perforator (DIEP) flap has become the most popular option for autologous breast reconstruction due to its assumed low donor site morbidity and natural aesthetic results [1].

It ideally represents the evolution of the transverse rectus abdominis myocutaneous flap (TRAM), with the significant advantage of sparing the rectus abdominis muscle with subsequent preservation of the abdominal wall integrity [2,3,4].

According to the Mathes and Nahai classification, the rectus abdominis muscle receives a type 3 vascular supply, with two dominant pedicles, one at each end of the muscle [5].

The last three posterior lumbar arteries and the deep circumflex artery can provide small anastomoses with the epigastric arteries’ lateral branches, thus contributing to rectus abdominis vascular supply [6].

In breast reconstruction with an ipsilateral DIEP flap, the internal mammary artery of the same side is the first choice for an anastomose with the deep inferior epigastric artery; only in selected cases do surgeons prefer the thoracodorsal vessels [7,8].

This way, even the superior epigastric artery, which is the anatomic extension of the internal mammary, is interrupted.

The interruption of both rectus muscle dominant pedicles can produce a dramatic reduction in the blood supply with consequent muscle belly ischemic alterations [9].

Under these circumstances, the secondary vascular network may develop hypertrophy similar to that commonly observed in other body areas [10], but its contribution remains uncertain.

The rectus abdominis muscle receives motor and sensory innervation from the T6 to T12 and L1 spinal nerves; all these nerves penetrate the rectus muscle belly at the level of its lateral side [11].

During surgical procedures, such as breast reconstruction with a DIEP flap, the surgeon should consider the risk of damaging motor innervation, which could lead to increased donor site morbidity [12].

The importance of avoiding iatrogenic denervation during DIEP flap dissection has already been noted by Rozen et al. [13], who suggested choosing medial perforators whenever possible to preserve the lateral portion of the motor unit.

Some authors have investigated the complications occurring in the abdominal wall after the DIEP flap harvest. None of these authors correlated any anatomical degenerative finding, instrumentally detected, with the possible variations in the strength and functionality of the muscle itself [14,15,16,17,18].

At present, no study has definitely stated if a reduced vascular supply together with motor nerve fiber interruption can compromise the rectus abdominis muscle composition and function.

For the first time, we report the correlation between imaging and clinical changes to the abdominal wall in patients who have undergone autologous breast reconstruction with a DIEP flap.

## 2. Materials and Methods

An observational, single-blind study was performed on a cohort of 12 women who had undergone mono- or bilateral breast reconstruction with a DIEP microsurgical flap, consecutively performed in our institute from 2013 to 2016 (Table 1).

The median age was 53.9 years, 3 patients had not been pregnant, only 1 had undergone caesarean section (2 births), 1 patient had a laparoscopic hysterectomy, and no other previous abdominal surgery was reported.

The cohort was divided into two groups: “ipsilateral reconstruction”, defined as reconstruction with a perforator flap based on the deep inferior epigastric artery (DIEA) on the same side of the affected breast; “contralateral reconstruction”, which indicates a flap harvested on the contralateral hemisome with respect to the affected breast.

Bilateral reconstructions are to be considered as two ipsilateral reconstructions, as both superior and inferior epigastric arteries are interrupted.

All patients were assessed by a preoperative computed tomography (CT) scan to identify perforator vessels, and by a postoperative CT scan scheduled for oncologic follow-up.

All CT scans were performed using a Siemens SOMATOM^®^ Definition 64-layer instrument, using the contrast agent Iomeron^®^ 400 (Bracco Imaging Italia s.r.l.).

The standard procedure provided a first scan without a contrast agent and a subsequent scan with a contrast agent in the arterial phase by bolus tracking, with an enhancement peak in the abdominal aorta at the renal site at 100 Hounsfield units (HUs) and acquisition after 4 s.

This was followed by the venous phase acquisition after 85 s from the start of the contrast agent injection.

The acquisitions were all reconstructed with 3 mm slices; MIP, VR, and MPR reconstructions were also performed.

All CT scans were assessed by a single radiologist, who compared the recti muscle bellies pre- and postoperatively, performing 2D measurements at different cross-section levels, detecting any changes in muscle structure and highlighting any postoperative modification of the abdominal wall.

Conventionally, the navel level represents the edge between the superior half muscle belly, mainly supplied by the superior epigastric artery, and the inferior one, supplied by the deep inferior artery.

Two prearranged CT scan slices were obtained for every single rectus muscle, passing in a transverse plane at an 8 cm distance proximally and distally, respectively, from the navel.

Derogation from this standard of measurement was allowed only when the preordered distance of 8 cm coincided with one of the fibrous bands, called tendon intersections, that divide the rectus muscle into separate bellies [19].

In that case, measurement and evaluation of the cross-sectional area were performed proximally and distally to the upper navel level and the lower navel level, respectively, in order to examine the largest section of the muscle belly.

Then, a comparison was made between the size of the muscle bellies before and after breast reconstruction surgery.

To evaluate any postoperative change in the muscle belly composition, a specific pre-established grading scale was used to obtain more objective results.

The grading scale was derived from soft tissue segmentation thresholds defined by specified HU values for fat, loose connective tissue or atrophic muscle, and normal muscle: intramuscular fat (HU −200 to −10), low-density muscle (HU −9 to 40), muscle (HU 41 to 70), and fibrous connective tissue (HU 71 to 150) [20,21].

Then, all patients were subjected to postoperative clinical and instrumental evaluation by a single physiatrist to correlate the CT findings with any postoperative movement impairment or pain.

Abdominal diastasis was studied using ultrasound and clinical evaluation to measure the distance between the recti muscles [22].

Spine rigidity was determined by measuring the minimum distance from the patient’s fingers to the floor while flexing the trunk and keeping the knees straight [23,24].

The Schober test was also used: the examiner traces a reference line passing through the spinous process of L5 and places a parallel line about 10 cm above the previous line with the patient standing. In normal subjects with maximum trunk flexion keeping their knees straight, the distance between the lines will increase more than 4 cm [25,26].

Recti muscle performance was assessed by a sit-up test: patients in a supine position were asked to keep their heels raised off the bed with their legs extended and to maintain this position for 10 s [27].

Possible algic symptoms due to scarring and back pain were investigated by administering a dedicated VAS scale.

Regarding disability due to low back pain, two questionnaires were submitted: the Oswestry Disability Index 2.1a-Italian version (ODI-I) [28,29], and the Roland Morris Disability Questionnaire-Italian version (RMDQ-I) [30,31].

Both investigations were conducted in a single-blind manner, in order to make the operator’s assessment more objective.

Neither the radiologist nor the physiatrist knew what type of procedure was performed (ipsi- or contralateral reconstruction), or the side of the rectus muscle donor site.

Patients who did not undergo a preoperative CT scan or refused physiatrist examination were excluded from the study.

The median postoperative follow-up with a CT scan and physiatrist tests was 27.75 months, with a range from 12 to 51 months.

## 3. Results

### 3.1. Two-Dimensional Evaluation

Table 2 reports the findings related to the CT cross-sectional measurements of rectus muscle bellies.

The CT scans were intended to be 8 cm above and 8 cm below the navel level. For any muscular belly investigated, the width and thickness were calculated, together with any variation (Δ-variation).

### 3.2. Muscle Quality Assessment

In agreement with the grading scale derived from soft tissue segmentation thresholds defined by the specified HUs, the assessment of the muscle density and composition in the preoperative and postoperative periods was scored as follows: 1, normal (no appreciable alterations); 2, mild alterations (less than 30% of the muscle cross-sectional area); 3, moderate alterations (from 30% to 50% of the muscle cross-sectional area); 4, severe alterations (≥50% of the muscle cross-sectional area).

Table 3 reports the evaluation of the change in the composition of rectus muscle bellies and the relative Δ-variation.

### 3.3. Physiatrist Investigation

#### 3.3.1. Physical Examination

Table 4 reports the summary of the postoperative physical tests, including US/manual measurement of diastasis recti abdominis, the fingertip distance to floor test, motion assessment of lumbar flexion, and the sit-up test.

#### 3.3.2. Pain and Impairment Assessment

Postoperative pain and disability outcomes are shown in Table 5.

VAS abdomen measures the intensity of pain related to the bisiliac sovrapubic scar after closure of the abdominal donor site, and VAS lower back measures the pain due to postoperative stiffness.

The ODI-I and RMQ-I values were derived from impairment caused by lumbar pain.

## 4. Discussion

Some studies suggest that the principal cause of abdominal wall postoperative bulge deformity is the lesion of the intercostal nerves which perforate the recti fascia posterior layer medially to the lateral row of perforators [9,10].

These findings prompted some surgeons to harvest DIEP flaps based exclusively on medial row perforators (M-DIEP) in order to reduce postoperative muscle wall impairment, despite a more tedious dissection compared to lateral row perforators (L-DIEP) [12,13].

Despite this hypothesis, there is no study in the literature focused on comparisons between pre- and post-imaging of the rectus muscle after DIEP flap harvest.

In our study, based on a comparison between preoperative and postoperative examination with computed tomography, a number of dimensional changes can be reported.

The two dimensions of the transverse area of the muscle can vary depending on different stress factors, and therefore some clinical considerations can be drawn.

The width, that is, the measurement of the transverse distance from the lateral to the medial muscle border, results were markedly lower in some patients (pt n# 2, 4, 6, 7, 11).

This could be associated with the wall repair after the dissection of perforators: usually, sutures tighten the muscle belly and fascia, so the transverse section shortens, as can be noted especially in the muscle sub-umbilical half (pt n# 4 to 7, and 9, 11, 12) (Figure 1).

Another interpretation is that, postoperatively, only the lateral portion of the muscle belly preserves contractility and a trophic composition, due to preservation of undisturbed lateral motor nerves.

Some of the muscle belly medial portion can start a process of atrophy and fibrotic degeneration, because the blood supply is significantly decreased; hence, the CT scan contrast agent cannot reach concentration levels.

Conversely, in some patients (pt n# 4, 5, 6, 12), the contralateral rectus seems to stretch towards the flap donor site; this could be due to abdominal wall progressive postoperative elongation due to the transverse line of tension (Figure 2).

The increase in both recti muscles’ width (pt n# 1, 8, 10) seems not to correlate with the side of the flap harvest but may indicate overall relaxation of the fascia, even if it does not appear to affect strength (Figure 3).

Muscle thickness, that is, the measurement of the distance between the anterior and posterior borders, seems to vary less after surgery; however, while the donor side width usually decreases, sometimes it can increase, probably due to compensatory hypertrophy.

Medial row perforators were largely preferred in this cohort (13 of 14 dissections, Table 1) due to vessel size, flap design, and lateral motor nerve preservation.

This type of surgical approach is frequently associated with a mild muscle thickness increase, as observed in patient n# 2, 5, 7, 8 (left side), 10, and 11 (Figure 4).

When not present at the muscle donor site, an increase in the contralateral rectus thickness may develop as a compensatory response (pt n# 1, 3, 4, 6) (Figure 5).

The use of computed tomography as a validated tool to assess intramuscular adipose tissue (IMAT) through muscle attenuation has already been reported [32].

Similarly, the between-muscle comparison was found to be effective [33].

In our study, all other patients seem to have maintained a normal muscle structure and composition despite transverse 2D changes.

Only patient n#3 showed postoperative marked involution (Table 3), and it seems to perfectly match with the result after physiatrist examination of the worst of the whole cohort (see Table 4 and Table 5), but apparently does not correlate with any cross-sectional area variation (Table 2).

This can be explained by preserving the function of the motor unit by choosing the row of medial perforators while harvesting the flap.

The integrity of the abdominal wall, recovered by suturing the fascia under tension, can be correlated with a deformation of the muscle belly without actually affecting its composition and quality.

Furthermore, the choice to harvest ipsilateral flaps was not found to be an independent factor in an increased risk of muscle attenuation, as shown in Table 4.

Probably, the many arborising branches of the intercostal and lumbar vessels can support the blood supply regardless of the contribution of the two main epigastric vessels.

These assumptions seem to reflect the results obtained by the physiatrist.

Physiatrist examination did not reveal recti muscle diastasis exceeding the physiologic grade; nine patients showed diastasis equal to or greater than 1.5 fingerbreadth.

Joueidi et al. found a substantial discordance among authors on what is the minimum distance of the recti that defines a diastasis. In our study, ultrasound measurement substantially confirmed clinical findings, and only two patients had a diastasis greater than 30 mm in width, which we set as the threshold value [34].

The Schober test did not evidence lumbar spine flexion impairment (0 of 12 patients). The flexion test instead resulted in an abnormal value in three patients; however, a formal bias is that the exam was conducted only postoperatively, and thus patients may have had some degree of spine stiffness even before surgery.

Similarly, the strength test evidence indicated severe impairment in three patients, but pre-surgery data were not available.

Physiatrists subjected the patients to the VAS scale for both low back pain and abdominal pain: three patients indicated abdominal pain greater than 30 in the past month, another two reported lower back pain greater than 30, and only one patient reported being affected by both abdominal and lower back pain.

The Oswestry Disability Index (ODI-I) indicated that eight patients had different degrees of disability: of them, six had a minimum grade, one had a moderate grade, and only one showed severe disability (pt n#3).

The Roland Morris Disability Questionnaire (RMDQ-I) revealed that some patients were affected by a disability: five had a low disability grade, and one patient showed a medium degree of disability (pt n#3).

No patient referred to practicing sports routinely, or to having changed their own daily activities (e.g., job, housework) after breast reconstruction with a DIEP flap, so their lifestyle was judged to be unmodified.

According to this survey, patient n#3 reported the highest levels of pain and the highest degree of disability and was the only one who required a second surgery for cosmetic correction of an asymptomatic inferior abdominal bulge deformity.

Interestingly, she was undergoing treatment for chronic depression.

## 5. Conclusions

Computed tomography shows that the muscle wall is constantly developing postoperative sequelae after harvesting the DIEP flap, even if these are not directly correlated with a functional impairment demonstrated by the physiatrist.

Leaving the lateral motor nerves intact certainly plays a role in the avoidance of severe deformation of the abdominal rectus muscle and major complications such as hernia or swelling.

The medial perforators are therefore the vessels of choice on which the DIEP flap harvest is based.

The highest grades of dimensional modification appear to happen to the rectus muscle in ipsilateral breast reconstruction, but a limited cohort study does not allow a correlation with specific disadvantages in terms of pain and motor impairment.

On the other hand, a wider physiatrist study could highlight some advantages in the contralateral DIEP procedure, especially if conducted before and after the operation.

This study also demonstrates that the section of both epigastric vessels does not match with any impairment of overall muscle blood supply, as proved by the substantially unmodified contrast agent uptake in postoperative scans.

## Figures and Tables

**Figure 1 diagnostics-12-00683-f001:**
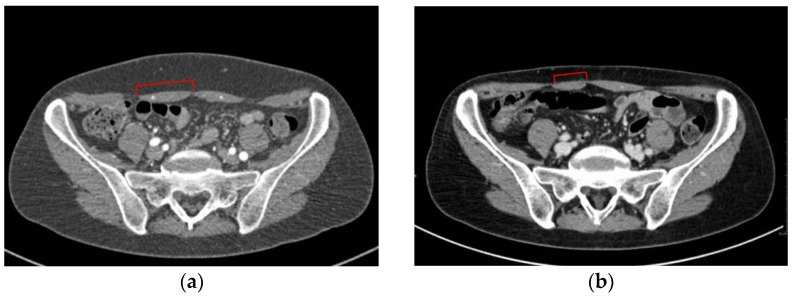
(**a**) Pt n# 11, preoperative transverse CT scan at sub-umbilical level; (**b**) pt n# 11, postoperative transverse CT scan at the same level. Red square brackets indicate the width of right rectus muscle belly (donor side of the DIEP flap).

**Figure 2 diagnostics-12-00683-f002:**
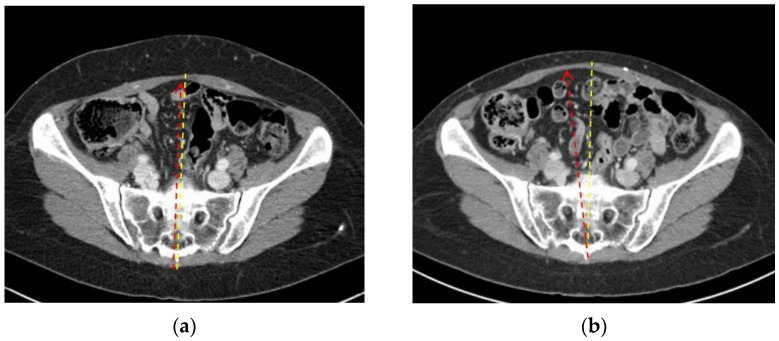
(**a**) Pt n#12, preoperative transverse CT scan at sub-umbilical level; (**b**) pt n#12, postoperative transverse CT scan at the same level. Yellow dotted lines indicate the midline, and red dotted arrows indicate the linea alba between the recti muscles.

**Figure 3 diagnostics-12-00683-f003:**
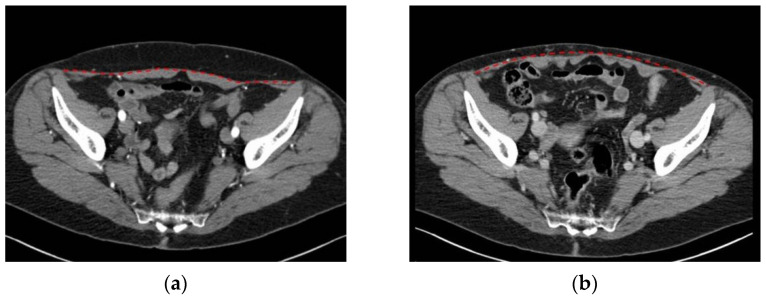
(**a**) Pt n#8, preoperative transverse CT scan at sub-umbilical level; (**b**) pt n#8, postoperative transverse CT scan at the same level. The red dotted line indicates the fascia above the recti muscles.

**Figure 4 diagnostics-12-00683-f004:**
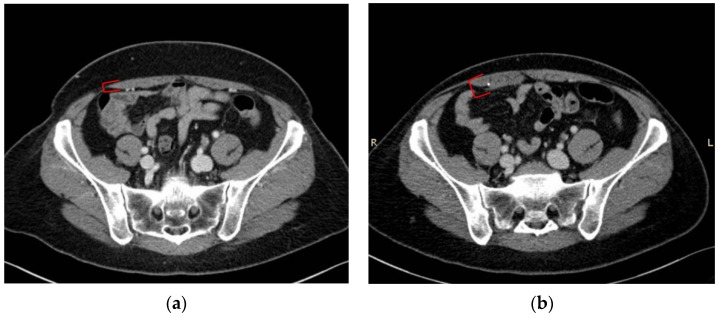
(**a**) Pt n#7, preoperative transverse CT scan at sub-umbilical level; (**b**) pt n#7, postoperative transverse CT scan at the same level. Red square brackets indicate the right rectus muscle thickness (donor site of the DIEP flap).

**Figure 5 diagnostics-12-00683-f005:**
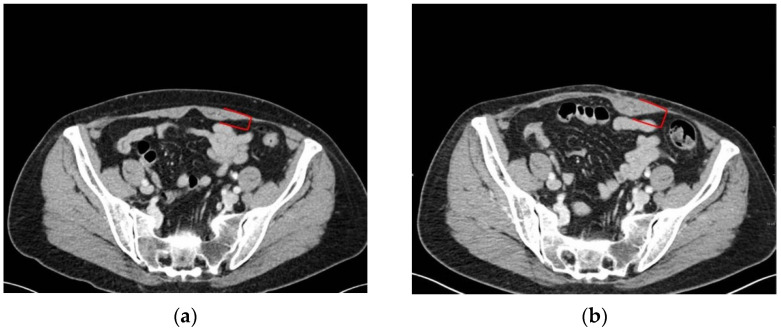
(**a**) Pt n#3, preoperative transverse CT scan at sub-umbilical level; (**b**) pt n#3, postoperative transverse CT scan at the same level. Red square brackets indicate the left rectus muscle thickness (contralateral site with respect to the donor side of the DIEP flap).

**Table 1 diagnostics-12-00683-t001:** Patient and flap characteristics.

Pt	Age	BMI *	Birth **	Comorbidity	Affected Breast	Donor Site ***	Perforator Vessels Side/Number/Row
1	54	34.4	2	hypertension, smoker	Left	Ipsilateral	Left/2/medial
2	50	30.9	2	sarcoidosis	Left	Ipsilateral	Left/2/medial
3	62	31.2	2	hyperthyroidism, depression	Left	Contralateral	Right/2/medial
4	61	34.4	3	diabetes, hypertension, hyperaldosteronism, dyslipidaemia	Right	Ipsilateral	Right/2/medial
5	45	35.8	1	none	Right	Ipsilateral	Right/1/medial
6	39	26.3	3	hypothyroidism	Right	Ipsilateral	Right/2/medial
7	52	29.7	2	hypercholesterolaemia	Right	Ipsilateral	Right/2/lateral
8	60	28.5	3	none	Bilateral	Bilateral	Right/2/medial Left/1/medial
9	56	30.4	none	none	Bilateral	Bilateral	Right/1/medial Left/2/medial
10	56	24.9	none	hypothyroidism	Right	Contralateral	Left/2/medial
11	60	18.2	none	hypertension, hypercholesterolaemia	Left	Contralateral	Right/2/medial
12	52	29.5	2	none	Left	Contralateral	Right/2/medial

* Body mass index in kg/m^2^; ** number of births (eventual); *** side of DIEP flap harvesting/rectus muscle dissection.

**Table 2 diagnostics-12-00683-t002:** Rectus abdominis muscle 2D variation.

Pt	Side *	Level **	Right Pre-Op		Right Post-Op		Right Δ ***		Left Pre-Op		Left Post-Op		Left Δ ***	
			Width	Thickness	Width	Thickness	Width	Thickness	Width	Thickness	Width	Thickness	Width	Thickness
1	Left Ipsi	above	82	7	84	8	2	1	87	7	91	8	3	1
		below	63	7	75	7	**12**	0	81	8	82	6	1	−2
2	Left Ipsi	above	97	11	81	10	**−16**	−1	109	8	82	10	**−27**	2
		below	84	12	70	12	**−14**	0	102	9	100	12	−2	**3**
3	Left Contra	above	78	10	72	10	−6	0	78	9	75	10	−3	1
		below	69	13	65	6	−4	**−7**	59	13	65	21	6	**8**
4	Right Ipsi	above	83	10	82	9	−1	−1	82	8	92	9	**10**	1
		below	69	14	41	14	**−28**	0	67	15	63	17	−4	2
5	Right Ipsi	above	74	11	73	13	−1	2	72	11	75	11	3	0
		below	72	15	63	16	−9	1	61	14	65	14	4	0
6	Right Ipsi	above	64	11	71	7	7	**−4**	64	9	69	7	5	−2
		below	50	11 m/10 l	62	7 m/11 l	**−12**	**−4 m/−1 l**	48	10 m/9 l	66	16 m/16 l	**18**	**6 m/7 l**
7	Right Ipsi	above	85	8	88	11	3	**3**	75	10	83	11	8	1
		below	78	11	63	15	**−15**	**4**	77	10	81	11	4	1
8	Bilateral	above	69	10	70	9	1	−1	64	8	75	9	9	1
		below	58	10 m/9 l	68	10 m/12 l	**10**	**0 m/−3 l**	58	11 m/10 l	69	11 m/14 l	**11**	**0 m/4 l**
9	Bilateral	above	58	11	62	8	4	**−3**	57	9	56	7	−1	−2
		below	57	9	54	10	−3	1	55	11	33	9	−2	−2
10	Right Contra	above	64	10	66	10	2	0	60	10	68	11	8	1
		below	45	4	47	6	2	2	48	14	51	15	3	1
11	Left Contra	above	45	9	38	10	−7	1	41	10	36	11	−5	1
		below	44	8	30	8	**−14**	0	43	8	51	9	8	1
12	Left Contra	above	66	11	65	10	−1	1	66	11	64	10	−2	−1
		below	80	11	74	9	−6	−2	67	11	73	11	6	0

Note: * side of flap harvest with respect to the affected breast; ** level with respect to the transverse plane passing through the umbilicus; *** Δ indicates variations between pre-op and post-op. When longitudinal rectus muscle schisis is present, measurements of each half are indicated (m = medial belly; l = lateral belly). Significant variations (Δ ≥ ±10 mm for width; Δ ≥ ±3 mm for thickness) are indicated in bold.

**Table 3 diagnostics-12-00683-t003:** Rectus muscle quality CT scan evaluation.

Pz	Breast/Rectus Side *	Level **	Right Pre-Op	Right Post-Op	Right Δ	Left Pre-Op	Left Post-Op	Left Δ
1	Left/Ipsilateral	above	1	1	0	1	1	0
		below	1	1	0	1	1	0
2	Left/Ipsilateral	above	1	1	0	1	1	0
		below	1	1	0	1	3	**+2**
3	Left/Contralateral	above	1	2	**+1**	2	2	0
		below	1	4	**+3**	1	1	0
4	Right/Ipsilateral	above	1	1	0	1	1	0
		below	1	1	0	1	1	0
5	Right/Ipsilateral	above	1	1	0	1	1	0
		below	2	2	0	2	2	0
6	Right/Ipsilateral	above	1	1	0	1	1	0
		below	1	1	0	1	1	0
7	Right/Ipsilateral	above	1	1	0	1	1	0
		below	1	1	0	1	1	0
8	Bilateral	above	3	3	0	2	2	0
		below	1	1	0	1	1	0
9	Bilateral	above	1	1	0	1	1	0
		below	1	1	0	1	1	0
10	Right/Contralateral	above	1	1	0	1	1	0
		below	4	4	0	1	1	0
11	Left/Contralateral	above	1	2	**+1**	1	1	0
		below	1	1	0	1	1	0
12	Left/Contralateral	above	1	1	0	1	1	0
		below	1	1	0	1	1	0

Note: * breast affected/donor side of flap; ** level with respect to the transverse plane passing through the umbilicus. Muscle density and composition evaluation criteria: 1, normal; 2, mild alterations; 3, moderate alterations; 4, severe alterations. Significant alterations (Δ ≥ ±1 pts) are in bold.

**Table 4 diagnostics-12-00683-t004:** Summary of physical examination.

Pt	Breast/Rectus Side	Diastasis (Fingers)	Diastasis (US)	Schober Test	Trunk Flexion	Strength Test
1	Left/Ipsilateral	3	**32**	0	0	10
2	Left/Ipsilateral	0	0	0	0	10
3	Left/Contralateral	2.5	**40**	0	0	**4**
4	Right/Ipsilateral	3	18	0	**9**	10
5	Right/Ipsilateral	3	20	0	**6**	10
6	Right/Ipsilateral	1	28	0	0	**4**
7	Right/Ipsilateral	1	0	0	0	10
8	Bilateral	2	24	0	0	10
9	Bilateral	1.5	16	0	0	10
10	Right/Contralateral	2	12	0	**14**	10
11	Left/Contralateral	2	22	0	0	**8**
12	Left/Contralateral	3	28	0	0	**3**

Note: diastasis was considered significant if greater than 30 mm; the Schober test was considered significant for any increase ≥4 cm; trunk flexion stiffness criteria: 1–4 cm = mild, 5–9 cm = moderate, ≥10 cm = severe; grading scale of strength impairment: ≥10 normal, 8–9 minimal, 5–7 moderate, 0–4 severe; significant values are in bold.

**Table 5 diagnostics-12-00683-t005:** Summary of survey response.

Pt	Breast/Rectus Side	VAS Abdomen	VAS Lower Back	ODI-I	RMDQ-I
1	Left/Ipsilateral	7	5	0	0
2	Left/Ipsilateral	13	7	6	1
3	Left/Contralateral	**79**	**100**	42	11
4	Right/Ipsilateral	**68**	4	28	6
5	Right/Ipsilateral	22	0	2	0
6	Right/Ipsilateral	3	1	0	0
7	Right/Ipsilateral	16	**51**	6	1
8	Bilateral	**96**	8	18	2
9	Bilateral	0	0	0	0
10	Right/Contralateral	0	**57**	12	0
11	Left/Contralateral	28	8	0	1
12	Left/Contralateral	**36**	0	8	0

Note: muscle VAS score for abdomen and lower back pain was considered significant for values ≥ 30; ODI-I disability thresholds: 0–20 minimum, 21–40 moderate, 41–60 severe, 81–100 complete; RMDQ lower back pain thresholds: 0–9 mild, 10–13 moderate, ≥14 severe; significant values are in bold.

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
