# Peer review of "A New CT Analysis of Abdominal Wall after DIEP Flap Harvesting"

_diagnostics, 2022, doi:10.3390/diagnostics12030683_

Round 1
Reviewer 1 Report
With the observation that abdominal microsurgical flap based on the DIEP perforator has become the most popular option worldwide for autologous breast reconstruction Brambullo et al. designed and implemented new diagnostic muscle imaging analysis. Overall, this study sheds light on developing new intervention approaches, however, some corrections are recommended for providing clear information. Particularly, I listed the following comments in detail here.
Major concerns:
In the abstract, the sentence must be written as past tense, please change “This study focuses on” to “This study focused on”, and “Patients are divided into” to “Patients were divided into”, and so on. Also, the finding of the assay could be added step by step based on methods. I recommend considering regular assays and results. All of the names and terms should be completely mentioned for the first time in abstract and text, for example, CT.
In the introduction, the citations of the literature are not appropriate, and some sentences lack reference. For example, “In breast reconstruction with ipsilateral DIEP flap, the deep inferior epigastric artery is generally anastomosed with the internal mammary artery of the same side”, “Rectus abdominis muscle receives motor and sensory innervation from T6 to T12 and L1 spinal nerves, all these nerves penetrate rectus muscle belly at level of its lateral side.”
In the discussion, discuss your results before relating them to the results of other published work. What is your conclusion? Do the authors have more thoughts on this field?
Author Response
Response to Reviewer 1 Comments
Point 1: In the abstract, the sentence must be written as past tense, please change “This study focuses on” to “This study focused on”, and “Patients are divided into” to “Patients were divided into”, and so on.
Response 1: The text has been revised and all senteces are now written as past tense.
Point 2: Also, the finding of the assay could be added step by step based on methods. I recommend considering regular assays and results.
Response 2: The MPDI template requires adherence to the following sections: introduction, methods and materials, results and so on. The step-by-step presentation of results may be more intuitive, but will not respect the division of subsections, by collocating results across methods.
Point 3: All of the names and terms should be completely mentioned for the first time in abstract and text, for example, CT.
Response 3: The text has been throughout revised
Point 4: In the introduction, the citations of the literature are not appropriate.
Response 4: In general, the "introduction" section may include references, since it can be appreciated in a number of articles published by Diagnostics. Given the sub-specialty of this study, we felt it was important to introduce dedicated bibliographic references.
Point 5: In the discussion, discuss your results before relating them to the results of other published work.
Response 5: The Discussion has been checked and revised
Point 6: What is your conclusion?
Response 6: Computed Tomography is a useful tool to assess the abdominal wall changes after DIEP flap harvest, which are proved to be more frequent than evidenced by usual clinical follow up. This study also showed that the section of the two epigastric vessels corresponded to no alteration of the overall muscular blood supply.
Point 7: Do the authors have more thoughts on this field?
Response 7: Some others bias can affect the clinical outcomes, such as the grade of aggressiveness of surgical dissection, the efforts made to preserve as much as possible the integrity of motor unit and the limitation to one/two single perforators in flap harvest. In addition, a detailed stratification of a larger population sample would significantly improve the accuracy of the results.
Reviewer 2 Report
For the first time, the authors report the correlation between imaging and clinical changes to the abdominal wall in patients who have undergone autologous breast reconstruction with a DIEP flap.This study focuses on rectus abdominal muscle morphology and function in a single series of 12 consecutive patients analysed before and after breast reconstruction with a microsurgical DIEP flap. Patients are divided into two groups "ipsilateral reconstruction" and "contralateral reconstruction", depending on the side of flap harvest and breast reconstruction, then evaluated by CT scans scheduled for tumor staging, and clinically examined by a physiatrist. The authors conclude that DIEP microsurgical breast reconstruction is a safe procedure without major complications at the donor site, even if long-term alterations of rectus muscle is a common finding. Overall an interesting study with long-term follow-up. Two comments: 1. Sample size is too small and hence a limitation 2. It woul be of interest to analyse also function analyses of muscle strenght, e.g. abdominal wall weakness.Author Response
Response to Reviewer 2 Comments
Point 1: Sample size is too small and hence a limitation
Response 1: Population sample is limited, so further investigations are needed before drawing definitive conclusions. Outside this limit, the design of this study was found to be effective in demonstrating frequent changes in muscle size and composition.
Point 2: It would be of interest to analyze also function analyses of muscle strength, e.g. abdominal wall weakness.
Response 2: Recti muscles strength performance was assessed by Sit-up test, patients in supine position were asked to keep their heels raised from the bed with their legs extended and maintain this position for 10 seconds. The findings are reported in Table 4.